# 2D Measurements of the Angle of the Vestibular Aqueduct Using CT Imaging

**DOI:** 10.3390/brainsci13010047

**Published:** 2022-12-26

**Authors:** Diane Jung, Nimesh Nagururu, Ferdinand Hui, Monica S. Pearl, John P. Carey, Bryan K. Ward

**Affiliations:** 1Department of Otolaryngology-Head and Neck Surgery, Johns Hopkins University School of Medicine, Baltimore, MD 21205, USA; 2Department of Radiology, Children’s National Hospital, Washington, DC 20010, USA

**Keywords:** Meniere’s disease, vestibular aqueduct, CT imaging

## Abstract

Recently, Bächinger et al. developed a software that measures the angle between the vestibular aqueduct proximal to the vestibule and the distal vestibular aqueduct on computed tomography (CT) scans and found differences in the vestibular aqueduct angle between the hypoplastic and degenerative categories of Meniere’s disease (MD). Hypoplastic radiological findings were associated with the development of bilateral MD and hypoplastic changes were not found outside of fetal temporal bones and individuals with MD. The purpose of this study is to examine how the software developed by Bächinger et al. performs when applied to a large dataset of adult patients with varied otologic diagnoses. Adult patients who underwent high resolution flat panel CT scans without intravenous contrast (*n* = 301) were retrospectively reviewed. Measurements of the angle of the vestibular aqueduct were made using the previously developed software tool. The tool could be applied to measure the vestibular aqueduct angle in most CT scans of the temporal bones (*n* = 572 ears, 95%). While the majority of ears fell within the normal range of <120 degrees (*n* = 462, 80%), fourteen ears (2.3%) in 13 patients were found to have vestibular aqueduct angles that meet criteria for hypoplastic MD (>140 degrees). Only one of the 13 patients had a diagnosis of MD and not in the ear in the hypoplastic category. An inconsistent pattern of other otologic diagnoses were found among the 13 individuals meeting criteria for hypoplastic MD. Although prior reports indicate the software has prognostic value in individuals with MD, these results suggest that the software may have lower positive predictive value when applied to a large population of individuals with varied otologic diagnoses.

## 1. Introduction

Meniere’s disease (MD) is a vestibular disorder of the inner ear. Its prevalence has been reported as high as 513/100,000 people in population based studies [1]. There is currently no diagnostic test available for Meniere’s disease, and like many other vestibular disorders it is diagnosed by a collection of clinical symptoms: episodic attacks of spinning vertigo, tinnitus, hearing loss, and aural pressure [2]. The timing of these symptoms and their combinations vary in patients with Meniere’s disease. The etiology of Meniere’s disease is unknown. 

Due to the heterogeneous presentation of symptoms in patients with Meniere’s disease, clinicians have attempted to categorize this disease into sub-groups since Prosper Meniere first presented cases in which pathology of inner ear appeared to cause vertigo [3]. These attempts have ranged from classifying patients by the timing of symptoms to pathology of the inner ear [4]. The vestibular aqueduct is an inner ear structure that connects the endolymphatic sac to the vestibule and has often been associated with Meniere’s disease [5,6]. Recently, two categories of Meniere’s disease—hypoplastic and degenerative—have been reported based on differences in pathology involving the vestibular aqueduct or endolymphatic sac and associated differences in clinical presentation. Meniere’s disease with hypoplastic pathology presents with a vestibular aqueduct that terminates prematurely in the operculum and is associated with bilateral Meniere’s disease and increased severity of inner ear swelling called endolymphatic hydrops. Meniere’s disease with degenerative pathology presents with a damaged epithelium of the endolymphatic sac and is associated with unilateral Meniere’s disease and an increased age of disease onset [7]. 

Many papers have attempted to characterize the vestibular aqueduct using CT imaging. Because an enlarged vestibular aqueduct is the most common congenital malformation associated with children who present with hearing loss, CT imaging has been used to measure the width of the vestibular aqueduct [8]. Rarer attempts to measure the angle of the vestibular aqueduct using CT imaging include cumbersome methods such as projecting the CT image on a large surface using an overhead projector and making measurements by hand [9]. In 2019 Bächinger et al. developed software to measure the angle between the portion of the vestibular aqueduct proximal to the vestibule and the distal vestibular aqueduct (which they labeled the angle of exit of the vestibular aqueduct, $a_{exit}$ or ATVA) on computed tomography (CT) scans aligned to the plane of the horizontal semicircular canal. This software is unique in its ability to measure the ATVA rapidly and conveniently using clinical imaging. 

Using this measured ATVA, the authors found differences between the hypoplastic and degenerative categories of Meniere’s disease and between fetal and adult temporal bones without Meniere’s Disease [10]. Patients with Meniere’s disease and a hypoplastic vestibular aqueduct and normal fetuses had ATVA > 140°. Patients with Meniere’s disease and degenerative changes and normal adults had angles < 120° [10]. They theorized that the steep angle in patients with Meniere’s disease with hypoplastic pathology indicated that this category of patients were congenitally predisposed to developing Meniere’s disease due to an underdeveloped vestibular aqueduct, while patients with degenerative Meniere’s disease who shared a similar range of vestibular aqueduct angles as normal adults were not congenitally predisposed to Meniere’s disease but developed Meniere’s disease in later life due to other factors [10]. 

Having established that categories of hypoplastic and degenerative Meniere’s disease can be diagnosed by a measured angle, they published additional work using the vestibular aqueduct as a diagnostic measurement to define hypoplastic and degenerative Meniere’s Disease categories and to form the basis for further study of those two groups [11]. They also found that hypoplastic MD patients have a higher prevalence of posterior semicircular canal dehiscence (PCD), potentially predisposing them to the development of a third mobile window syndrome [12]. The software promises to be helpful for predicting outcomes in patients with MD, but has not been applied to a large dataset of temporal bones from individuals with a variety of otologic diagnoses. We wondered if patients with ATVA > 140° were unique to patients with MD or whether this observation could be found in other diagnoses. In particular, superior semicircular canal dehiscence syndrome (SCDS) is third mobile window syndrome thought to have a congenital origin in which patients can experience symptoms such as sound and/or pressure-induced vertigo, autophony, aural fullness and hyperacusis for bone conducted sounds, among other symptoms [13]. The purpose of this study is to examine how the software developed by Bächinger et al. performs in a large data set of high resolution CT scans of the temporal bone performed for a variety of otologic diagnoses.

## 2. Materials and Methods

### 2.1. Ethics

This study was approved by the Johns Hopkins Institutional Review Board (IRB ID: IRB00279939; Baltimore, Maryland).

### 2.2. Temporal Bone CT Imaging Data from Clinical Patients

Adult patients who underwent high resolution flat panel CT (DynaCT; Siemens, Erlangen, Germany) scans without intravenous contrast (*n* = 301, Table 1) at the Johns Hopkins Hospital from July 2018 through April 2021 were retrospectively reviewed. This flat panel technique yields near-isotopic voxels that permit detailed reconstructions in any plane with high spatial resolution, as small as 0.1 mm for bony structures. Imaging protocols were previously published [14]. The most common indication for the flat panel CT was evaluation for suspected dehiscence of the superior semicircular canal. Imaging studies were included if the patient was at least 18 years old. Commercially available software (Carestream PACS, v. 12.2.6.304, 2018, Amsterdam, The Netherlands) was used to view the images. Images were initially viewed in the axial plane. If the plane of the axial section was not aligned with that of the lateral semicircular canal, we reconstructed the image in the plane of the lateral semicircular canal for consistent measurements. Clinical data were accessed through an electronic medical record. Patients were identified as having a superior semicircular canal dehiscence or additionally as meeting the diagnostic criteria for superior semicircular canal dehiscence syndrome (SCDS) according to the international classification of vestibular disorders (ICVD) [13]. The radiologist identified an anatomic dehiscence by reformatting the images in the plane of and orthogonal to the superior semicircular canal and by looking for an absence of bone anywhere along the arc of the superior semicircular canal. For analysis, patients in the superior semicircular canal dehiscence (SCD) group had an anatomic dehiscence on CT imaging but did not meet the ICVD criteria for SCDS. Patients were additionally identified as having received a diagnosis of Meniere’s Disease per a review of their medical records. Patients who did not receive a diagnosis of Meniere’s Disease or radiologically identified dehiscence were classified in a control group for comparison to the MD, dehiscence only, and SCDS groups. Ears were analyzed by diagnosis given to patient. For example, even if a patient had a diagnosis of Meniere’s disease in only one ear, both ears were categorized as from a patient with Meniere’s disease. We analyzed the data in this way as both Meniere’s Disease and anatomic dehiscence of the superior semicircular canal can occur bilaterally in patients [15,16]. There was no overlap of patients between groups.

### 2.3. Vestibular Aqueduct Angle Measurements in CT Sections

All measurements of the vestibular aqueduct were made in the plane aligned with the horizontal semicircular canal. Software developed by Bächinger et al. for measurements of the angle of the vestibular aqueduct was used [1]. As described in Bächinger et al., a predefined shape (magenta shape, Figure 1) was manually fitted to the vestibule and horizontal semicircular canal. As it is difficult to visualize the entrance of the vestibular aqueduct to the vestibule in most CT scans and therefore measure the proximal angle $a_{entrance}$ of the vestibular aqueduct, Bächinger et al. predetermined Line L1 (red line, Figure 1) based on histologic measurements and fixed it to the magenta shape at 14 degrees from the medial wall of the vestibule. Line L2 (green line, Figure 1) was manually adjusted to run parallel to the distal vestibular aqueduct near the operculum. The angle $a_{exit}$ measured is the angle between L1 and L2 (Figure 1). Angles were assessed by author DJ and the first ten measurements were reviewed with the senior author BW. Measured angles were compared to the angles reported as thresholds in the Bächinger et al. publication [10]. The medical records for all patients were reviewed for a clinical diagnosis. Diagnosis of Meniere’s Disease was determined using the consensus criteria for the diagnosis of definite Meniere’s Disease as set by the Barany Society’s ICVD [17]. The diagnosis is based on 3 clinical symptoms: episodic vertigo, fluctuating aural symptoms (hearing, tinnitus, aural fullness), and low frequency sensorineural hearing loss [17].

### 2.4. Statistical Analysis

Values are reported as absolute numbers or medians with interquartile ranges. Values for the angle were not normally distributed as determined by the Shapiro–Wilk test (*p* < 0.05). Group comparisons among “Controls”, “SCD”, “SCDS” and “MD” were performed using Kruskal–Wallis test statistics. For group demographic comparisons and outlier frequency comparisons among study groups, Fisher’s exact test was performed. To assess interrater reliability of measurements, an intraclass correlation coefficients (ICC) using the “two-way random effects” model, type “single”, and definition “absolute agreement” was calculated for measurements performed by authors DJ, NN, and BW on a random sample of 20 ears. Calculated ICC for the sample was moderate according to Koo criteria (0.54, 95% CI 0.28–0.76) [18]. A *p*-value of 0.05 was used to determine statistical significance. R was used for all statistical tests (R Foundation for Statistical Computing, Vienna, Austria). 

## 3. Results

### 3.1. Demographics

Table 1 shows basic demographic information for the study population. In total 301 patients (572 ears) were included in this study. Thirty ears (5.0%) from 30 patients were excluded because the distal vestibular aqueduct in the region of the operculum was unable to be visualized. All imaging studies had an associated otologic diagnosis. Most ears (*n* = 407, 71.2%) did not have a clinical diagnosis of Meniere’s disease or dehiscence. One hundred and forty-four ears (25.2%) were from patients who were diagnosed with at least an anatomical dehiscence of the superior semicircular canal, with most of these ears (*n* = 112, 77.8%) from patients who met the clinical diagnosis of SCDS. Twenty-one ears (3.7%) were from patients who had received a clinical diagnosis of definite Meniere’s Disease. There were no significant differences among groups with respect to age (*p* = 0.397; however, there were differences in sex and race among groups (*p* < 0.05). Pair wise comparisons found a larger proportion of males in the group with “Meniere’s Disease” compared to “Control” and “SCDS” groups. Significant differences in race were found between “SCDS” and “Control” groups (Table 1).

### 3.2. Angle of the Vestibular Aqueduct (ATVA)

In this study 462 (80.8%) ears had angles $a_{exit}$ less than 120 degrees, 96 (16.8%) ears had angles 120 < $a_{exit}$ > 140 degrees, and 14 (2.4%) ears had angles greater than 140 degrees (Table 2). All measurements for ears with angles > 140 degrees are shown in Figure 2. There were no differences among the groups in the ATVA (Figure 3). There were 13 adult patients (4.3%) with $a_{exit}$ > 140 degrees. One patient had ears with $a_{exit}$ > 140 degrees bilaterally. One patient with $a_{exit}$ > 140 degrees had a clinical diagnosis of Meniere’s disease but not in the ear with $a_{exit}$ > 140. The medical records of all individuals meeting the criteria for hypoplastic vestibular aqueduct (>140 degrees) were additionally reviewed. None of the 13 patients had a clinical diagnosis of Meniere’s Disease in the affected ear. These patients were diagnosed with a variety of otologic problems. Of the 14 ears, diagnoses included superior canal dehiscence syndrome (*n* = 3), thin bone over the superior semicircular canal (*n* = 1), intracochlear schwannoma (*n* = 1), otosclerosis (*n* = 1), or asymptomatic/presbycusis without associated symptoms of Meniere’s disease (*n* = 7). Images were reformatted in the planes of both the superior SCC and the posterior SCC and orthogonal to these planes. For the affected ears, imaging was interpreted by the radiologist as normal (*n* = 8), to have a dehiscent superior SCC (*n* = 3) to have thin bone overlying the superior (*n* = 1) or posterior (*n* = 2) semicircular canals. Other incidental findings include a prior transverse sinus stent (*n* = 1), bony exostoses of the external auditory canal (*n* = 1), and a prior right mastoidectomy performed for superior canal plugging (*n* = 1). None of the patients had a dehiscence of the posterior SCC. One ear had episodic aural fullness and two episodes of spinning vertigo, but with a high frequency hearing loss on pure tone audiometry, meeting the diagnosis of probable Meniere’s disease.

### 3.3. ATVA: SCD vs. SCDS

The $a_{exit}$ among patients with a radiologically identified superior canal dehiscence, patients diagnosed with superior canal dehiscence syndrome, and patients diagnosed with Meniere’s Disease were compared. Median and interquartile range values for $a_{exit}$ were not significantly different among the three groups (Patients without MD or dehiscence: 108 (17.4) degrees, *n* = 407; Superior Canal Dehiscence: 105 (14.3) degrees, *n* = 32; Superior Canal Dehiscence Syndrome = 105 (16.8) degrees *n* = 112; Meniere’s Disease = 107 (18.6) degrees, *n* = 21) (Figure 3). Pairwise comparisons were therefore not performed.

## 4. Discussion

The aim of this study was to determine the distribution of vestibular aqueduct angles in a large population of individuals with a variety of otologic diagnoses and to examine whether other inner ear disorders like SCDS may have a hypoplastic vestibular aqueduct. Nearly all individuals in this study fell within the normal range for the angle of the vestibular aqueduct and 11 had a diagnosis of clinical Meniere’s disease. None of the affected ears of patients with Meniere’s disease in this study met the criteria for a hypoplastic vestibular aqueduct. Furthermore, no significant differences were found in the vestibular aqueduct angles among the patients with Meniere’s disease, patients with solely an anatomic superior canal dehiscence diagnosed on imaging, and patients with superior canal dehiscence syndrome. Among all CT scans, we found 13 patients (14 ears) with a vestibular aqueduct that exceeded the 140 degrees angle cutoff set by Bächinger et al. for patients with hypoplastic Meniere’s disease pathology, yet none of those patients had Meniere’s disease in the affected ear (Figure 3). One of the ears with ATVA > 140 degrees was from a patient with a diagnosis of Meniere’s Disease, but that diagnosis applied to the contralateral ear. Three of the ears with $a_{exit}$ > 140 had a diagnosis of SCDS, and the other 9 ears with $a_{exit}$ > 140 came from patients without Meniere’s disease or dehiscence of the superior semicircular canal. Thin bone was identified over the posterior semicircular canal in two cases. There did not appear to be a consistent pathology identified in individuals with a hypoplastic vestibular aqueduct on CT imaging.

Furthermore, CT datasets for 96 temporal bones had no identified anatomic abnormalities and had an ATVA between 120 and 140 degrees. These CT datasets came from individuals who did not have a diagnosis of Meniere’s disease, nor did they have symptoms that were attributable to other pathologic processes such as vestibular migraine or SCDS. These results suggest that although generally effective at identifying a normal angle of the vestibular aqueduct, 2D measurements of the vestibular aqueduct may be limited as some normal individuals in our dataset without a diagnosis of MD meet ATVA criteria established by Bachinger for hypoplastic MD.

These results may be due to limitations inherent to the software and use of a 2D image. Because the software involves manual and subjective adjustment to the predefined shapes and lines (Figure 1) used to measure the $a_{exit}$ of the vestibular aqueduct, there was variability of results depending on how the user manually fit the shapes and lines to the image. This is supported by our finding of moderate reliability in the measurements when the software is used by three different raters to evaluate a randomly selected subset of our sample. As presented in Figure 4, slight variations in how the user fit the predefined horizontal semicircular canal and vestibule shape (purple, Figure 4) to the same image resulted in an $a_{exit}$ that either fell below 120 degrees at 114.64 degrees and therefore in the category of normal adults or patients with degenerative Meniere’s pathology, or above 120 degrees at 132.47 degrees. This example would not be categorized as hypoplastic Meniere’s disease with $a_{exit}$ > 140; however, the variability places the patient outside of the established range for normal patients ($a_{exit}$ < 120). The algorithm could be further standardized with added constraints to improve the reliability of the measurements.

Additionally, in 30 patients, the distal vestibular aqueduct in the region of the operculum of one ear was determined too difficult to clearly visualize and measure when reformatted in the plane of the horizontal semicircular canal, even in these high resolution CT images. Figure 5 shows a patient whose distal vestibular aqueduct was unidentifiable in the image stack. Figure 6A–D demonstrates how although the image was captured in the plane of the horizontal semicircular canal, the distal portion of the vestibular aqueduct is not visible and takes a complicated trajectory through the bone. In addition, often the vestibular aqueduct is visualized as a small wedge-shaped figure (Figure 6A–D) that may be a representation of the vestibular aqueduct directly parallel to its entering the temporal bone rather than a visualization of the vestibular aqueduct perpendicular to its travel through the bone. This is partially accounted for by the software as there is an option to use both an image at the level of the vestibule and if the distal aqueduct is not visible in that image, another image at the level of the distal VA to measure $a_{exit}$.

Excluding limitations inherent to the software and imaging methods, differences in populations between our study and the Bächinger study may have produced ATVA > 140 degrees in patients without a diagnosis of Meniere’s disease. Patients from the Bächinger study were recruited from the Massachusetts Eye and Ear Infirmary in Boston, MA while our patients were recruited from the Johns Hopkins Hospital in Baltimore, MD. Regional differences in the local populations may contribute to anatomic differences between datasets. This study population also was enriched with patients with SCDS, who often travel from outside the Baltimore area for evaluation. In addition, our study is limited by a lack of a large group of individuals with clinical Meniere’s disease. The performance of the software may have declined as a result of the different study populations. Based on the data here, the tool has uncertain positive predictive value when applied to patients in whom additional data is needed to establish a diagnosis of Meniere’s disease. Despite these potential limitations with the software, most ears in those without clinical Meniere’s disease measured within the normal adult category ($a_{exit}$ < 120 degrees) established by Bächinger et al. Additionally, it is possible that the tool may have meaningful ability to predict bilateral MD in individuals who already have a diagnosis of Meniere’s disease as has been previously shown by Bächinger et al. [11].

At our institution, CT imaging is rarely performed for individuals with MD, but magnetic resonance imaging (MRI) is often performed in individuals with MD due to asymmetry in hearing measured by pure tone thresholds. Increasingly studies are using high-resolution MRI to assess the vestibular aqueduct, and Bächinger et al. have adopted their technique for MRI [12,19,20]. One recent study found 3D reconstruction of the vestibular aqueduct to be a more accurate assessment of vestibular aqueduct volume compared to 2D measurements [21]. While the spatial resolution of MRI currently lags that of CT, clinicians are likely to adopt the tool more readily for use in MRI. 

## 5. Conclusions

In conclusion, the tool developed by Bächinger et al. is a free, convenient, and easy to use instrument that could be improved by increased constraints placed on its application. Most individuals with otologic complaints had a normal angle of the vestibular aqueduct; however, this study found that measuring the angle of the vestibular aqueduct using 2D images to determine whether the vestibular aqueduct is hypoplastic has limited positive predictive value. Furthermore, we did not find consistent pathology or other imaging observations among the outlier cases in which an increased angle of the vestibular aqueduct was found. New technologic advances within 3D reconstruction have resulted in simpler methods of reconstruction from 2D data compared to initial efforts in the 1990s, and may better account for the complicated trajectory of the aqueduct [22,23].

## Figures and Tables

**Figure 1 brainsci-13-00047-f001:**
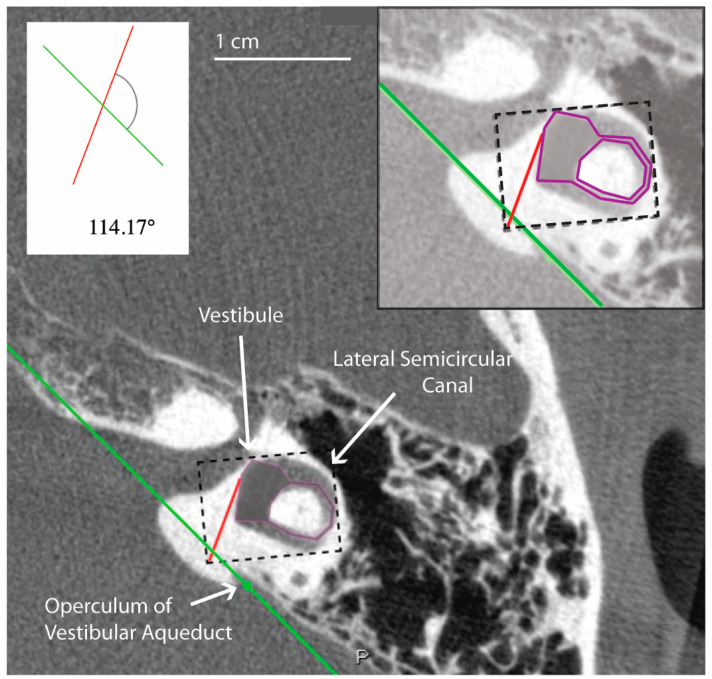
Example of temporal bone reconstruction from flat-panel CT imaging data. Ear (4L) measured using the tool developed by Bächinger et al. [1]. The magenta predefined shape is fitted to the vestibule and horizontal semicircular canal by altering the dimensions of the dashed rectangle. A red line is predefined by Bächinger et al. and fixed to magenta shape [1]. The green line is manually aligned parallel to the trajectory of the distal vestibular aqueduct. The inset shows a magnified view of the components used in the calculation of the angle, which in this case is 114.17 degrees and falls within the normal range.

**Figure 2 brainsci-13-00047-f002:**
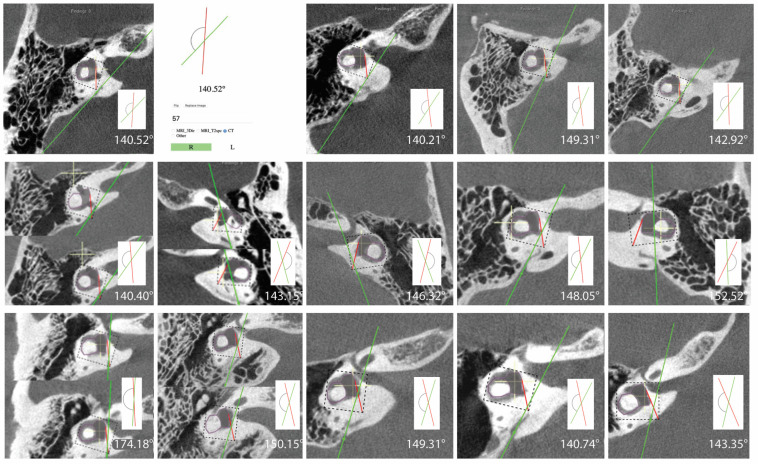
Images and measurements for all ears with ATVA > 140 degrees are shown. Bächinger et al. software used to measure the ATVA [1]. Some ears use two images to calculate the angle. If the vestibular aqueduct and the vestibule are not visible in a single plane, the software provides an option to use both an image at the level of the vestibule and another image at the level of the distal VA to measure $a_{exit}$.

**Figure 3 brainsci-13-00047-f003:**
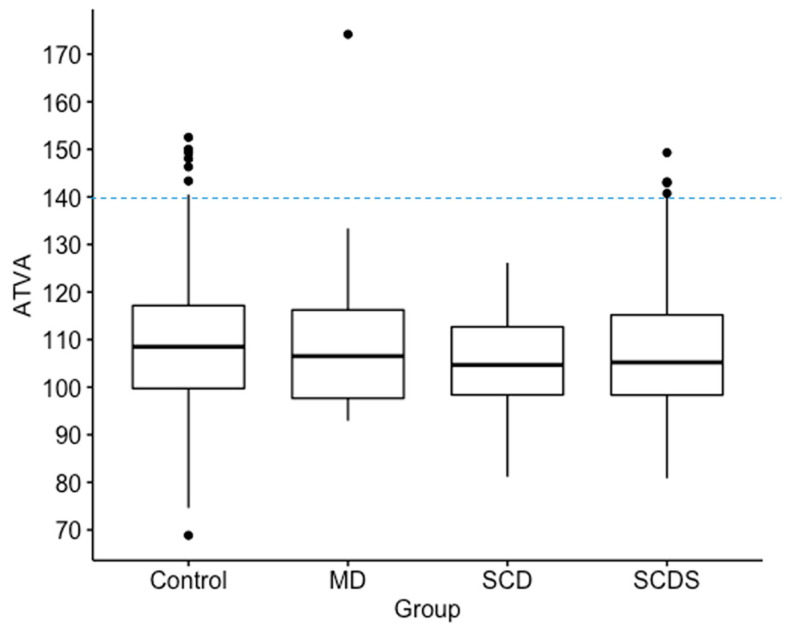
Box plots show median, interquartile range and outliers for angle of the vestibular aqueduct (ATVA). A cut-off line is shown at 140 degrees for defining abnormal as previously defined. No significant differences in $a_{exit}$ of the vestibular aqueduct was found among Meniere’s Disease (MD), superior canal dehiscence (SCD), superior canal dehiscence syndrome (SCDS), and control groups. In this figure, “Control” refers to patients without diagnosis of MD, SCD, or SCDS.

**Figure 4 brainsci-13-00047-f004:**
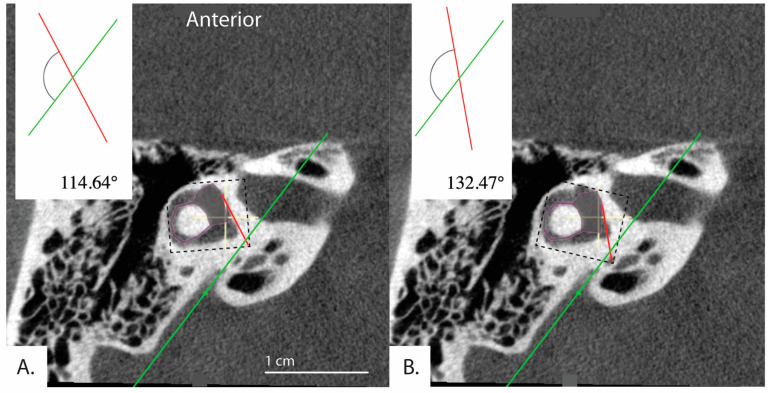
Example demonstrating variability in the determined angle depending on application of the tool. Bächinger et al. software used to measure angles in the same image. (**A**) $a_{exit}$ = 114.64° and (**B**) $a_{exit}$ = 132.47°. Depending on the settings of the software, different angles for the same image can result.

**Figure 5 brainsci-13-00047-f005:**
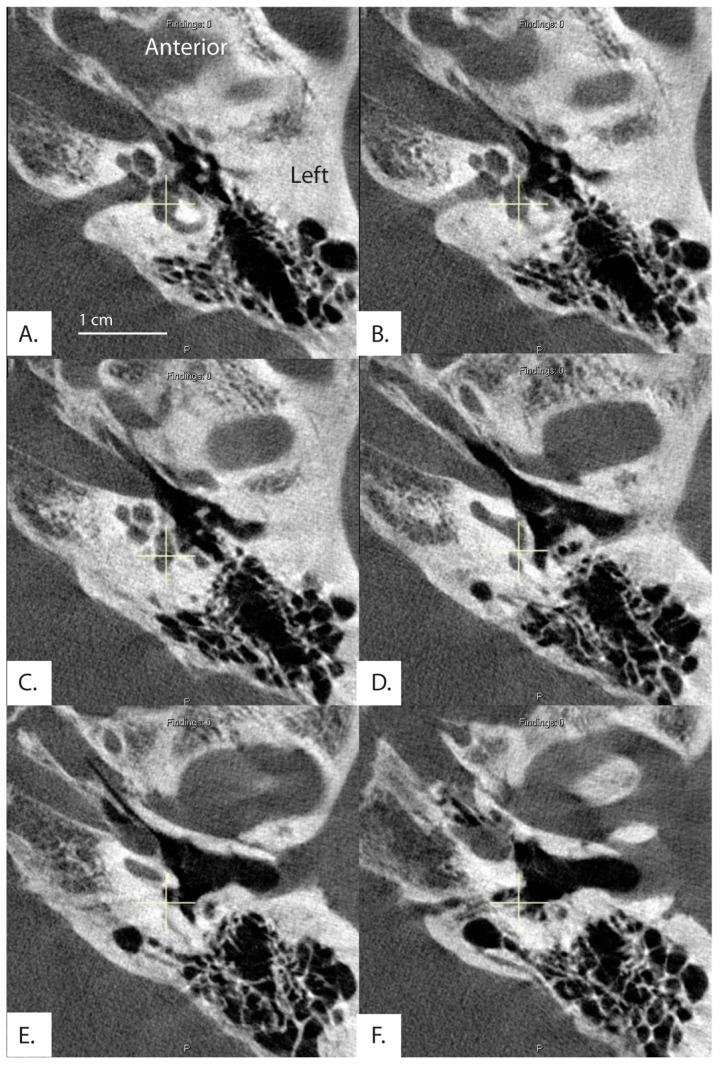
Image formatted in the plane of the horizontal semicircular canal. Images progress superior to inferior (**A**–**F**). Exit of the vestibular aqueduct in the opercular area of this patient (369L) was difficult to visualize and therefore measure in both ears.

**Figure 6 brainsci-13-00047-f006:**
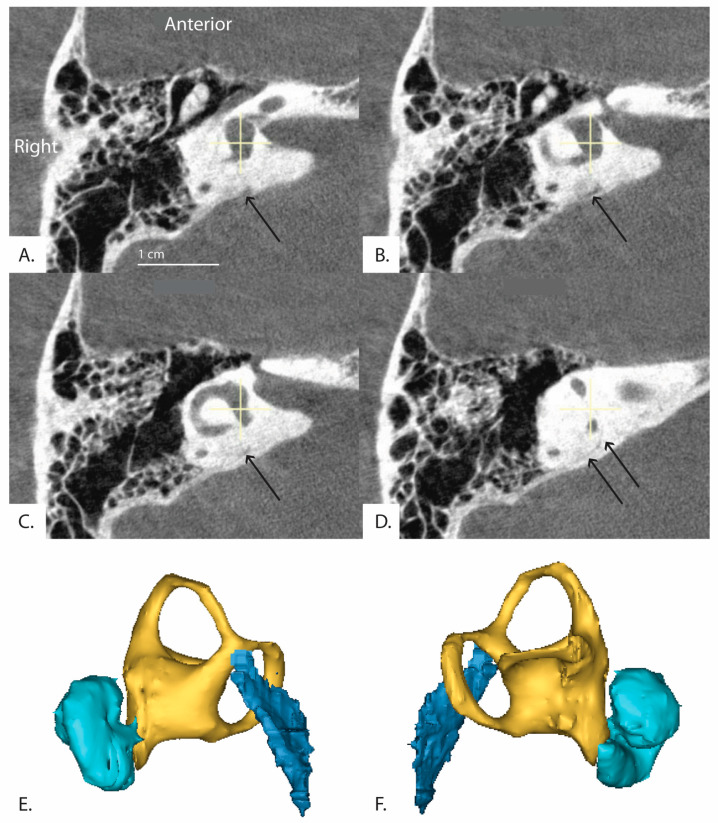
(Patient 16R) A series of sequential CT images of the inner ear reformatted in the plane of the horizontal semicircular canal are shown. The series runs clockwise from top left (**A**–**D**) and progresses inferior to superior. The distal vestibular aqueduct is unable to be visualized at the level of the vestibule but is shown in the image stack. Black arrows indicate the path of the vestibular aqueduct. (**E**,**F**) demonstrate 3D reconstruction of the vestibular aqueduct. (**E**) is a right view of the 3D model and (**F**) is a left view. Cyan colored shape is the cochlea, yellow is the vestibule and semicircular canals, and dark blue is the vestibular aqueduct.

**Table 1 brainsci-13-00047-t001:** Baseline Demographics.

	Total Patients	Control*	Superior Canal Dehiscence	Superior Canal Dehiscence Syndrome	Meniere’s Disease
*n*	301	212	17	61	11
Age at Time of Imaging (IQR)	50 (15.2)	49 (23.2)	59 (15.0)	50 (17.0)	51 (16.5)
Sex					
Males	116 (38.5%)	78 (36.8%)	9 (52.9%),	21 (34.4%)	8 (72.7%)
Females	185 (61.5%)	134 (63.2%)	8 (47.1%)	40 (65.6%)	3 (27.3%)
Self-Reported Race					
Caucasian	254 (84.4%)	169 (79.7%)	15 (88.2%)	60 (98.4%)	10 (90.9%)
Black	23 (7.6%)	20 (9.4%)	2 (11.8%)	1 (1.6%)	0
Asian	9 (3.0%)	9 (4.2%)	0	0	0
Other	12 (4.0%)	12 (5.7%)	0	0	0
Did not Identify	3 (1.0%)	2 (0.9%)	0	0	1 (9.1%)

Control* refers to patients without MD, or anatomic dehiscence of the superior canal.

**Table 2 brainsci-13-00047-t002:** ATVA* categorized by Bächinger criteria.

ATVA* in Degrees	Numbers of Ears (%)
X < 120	462 (80.8%)
120 < X < 140	96 (16.8%)
X > 140	14 (2.4%)

ATVA*: Angle of the vestibular aqueduct.

## Data Availability

The data presented in this study are available on request from the corresponding author. The data are not publicly available due to the presence of protected health information.

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
