# Peer review of "2D Measurements of the Angle of the Vestibular Aqueduct Using CT Imaging"

_brainsci, 2022, doi:10.3390/brainsci13010047_

Round 1

Reviewer 1 Report

Major comments:

It’s not clear in introduction why Bachinger’s software is used to measure CT scans of 77 the temporal bone performed for a variety of otologic diagnoses.

Most importantly, how did the authors verify Bachinger’s software’s performance regarding the angle measurements across the dataset in this work?

It’s not clear whether the focus of this research is the validation of the software, or the characterized results measured with the software, i., these results suggest that in 25 a large population of individuals with otologic disease, most do not have a hypoplastic vestibular 26 aqueduct.?

Other comments:

Line 9, line 53, developed “a” software

Not too common to see references in ABSTRACT.

Would like to see a broader introduction section, is the Bachinger software the only one suitable for this work?

Resolution of Figure 2 is too low.

Line 218, Look like a typo here? What is “Figure 3. Pairwise comparisons were therefore not per-218 formed”?

Reviewer 2 Report

This study evaluated the angle of the vestibular aqueduct using Bächinger's software in clinical patients and found no significant differences between Superior semi circular canal dehiscence syndrome (SCDS) / SCD / Meniere’s
Disease / Control. The results of this study is mainly a basic measurent theme and not so novel, whereas the content may be valuable to future clinical researchers. Following are the minor points to be fixed.
1) Was there any anomalies in the CT data set or is there any criteria to select control CT data set?
2) ... n=301 ... at the Johns Hopkins 88 Hospital from July, 2018 through April, 2021 ...
The reviewer do not know the size of the hospital, but the 100 cases per year seems to be small. Is there any criteria to be stated in the selection of cases? What is the total number of CT data set?
3) The diagnosis criteria of MD can be briefly stated somewhere.
4) When considering the circulation of the lympatics, the function of cochlear aqueduct was also described in the previous studies. Is it possible to evaluate the cochlear aqueduct in this software, or can be described in the discussion section.

Round 2

Reviewer 1 Report

The authors addressed most of my concerns. The quality of Figure 2 should be improved before publication.